# Snowmobiling and Climate Change: Exploring Shifts in Snowmobile Activity Using a Temporal Analogue Approach in Ontario (Canada)

Michelle Rutty * , Francesca Cardwell  and Grant Gunn 

Department of Geography and Environmental Management, University of Waterloo, Waterloo, ON N2L 3G1, Canada; fcardwell@uwaterloo.ca (F.C.); g2gunn@uwaterloo.ca (G.G.)
* Correspondence: michelle.rutty@uwaterloo.ca

**Abstract:** The multi-billion-dollar snowmobile industry is predicated on natural snowfall and cold temperatures, with a near absence of research that examines industry response to climatic variability and change. Using a temporal analogue approach, this study examines 30 years of climate data (1989–2019), along with operational (grooming hours) and performance (permit sales) indicators, to provide insight into the vulnerability and adaptive capacity of the Ontario snowmobile industry in a medium (RCP4.5) and high (RCP8.5) mid-century (2046–2060) emission scenario. The results underscore important temporal and spatial variability across Ontario's 16 snowmobile districts, indicating that snowmobilers are highly resilient to marginal conditions, changing districts and switching from seasonal to daily permits in response to warming temperatures. The findings from this study can inform risk assessments in other major snowmobile markets (e.g., Canada, Europe, USA), with future research needs discussed.

**Keywords:** snowmobiling; tourism; climate change; analogue; weather; behaviour

## 1. Introduction

Outdoor winter tourism and recreational activities are reliant on low temperatures and snowfall conditions, with wide agreement that inter-annual climatic variability impacts, and will continue to impact, the length and quality of winter seasons (i.e., rising temperatures, reduced natural snowfall) [1]. To date, the impact of climate change on winter tourism has predominantly focused on the ski industry [2,3], which highlights the indispensable role of snowmaking to offset marginal winter conditions (e.g., a shortened snow season, reduced depth of snowpack, poor quality snow) and enhance destination competitiveness [4,5]. However, not all snow-dependent tourism activities have the adaptive capacity to lengthen the winter season via snowmaking, snow farming, or artificial surfaces [6]. The impact of climate change on the multi-billion-dollar snowmobile industry has largely been overlooked, with a dearth of available research [7,8]. Snowmobile trails are often located at lower (i.e., warmer) elevations, covering thousands of kilometers across a large area of terrain, making the widespread implementation of snowmaking impractical [9]. As temperatures rise and natural snowfall decreases, the future sustainability of large snowmobile trail networks is uncertain.

Even though all major winter tourism markets have faced several record warm winters over the past two decades, researchers have not seized the opportunity to assess how operators and tourists have responded to marginal conditions [10]. Climate analogues (i.e., record climatic conditions that are representative of future normal conditions) enable researchers to empirically capture and assess both climatic vulnerability and adaptive capacity [11] to understand how climate and society interact [12]. Since climate alone does not determine the response of tourism stakeholders, an exploration of past events can shed light on the sensitivity of operators and tourists to climatic stimuli [13]). Analogue

studies are also particularly helpful within a tourism sector context, as the findings are more tangible and consistent with business investment norms versus hypothetical scenarios decades into the future [14]. Despite the value of an analogue approach to assess climate impacts across key tourism markets [15], research remains limited to a handful of studies, including ski [14,16], parks [17], and golf [18] tourism.

In Ontario (Canada), snowmobiling is recognized and celebrated as a premier winter recreation and tourism experience. In 2019, the Ontario snowmobile industry generated over $3.3 billion in total economic activity across its 34,000 km trail network (Ontario Federation of Snowmobile Clubs [OFSC] 2019). Nearly two decades ago, ref. [19] projected that climate change would eradicate snowmobiling in Ontario by the 2050s, but the model was not validated using industry data, nor did it differentiate impacts across the extensive snowmobile network. It therefore remains unclear whether the modelled projections overestimate the impact of climate change on snowmobiling, which has been a noted criticism in the ski tourism literature (e.g., [10,20,21]). Using a temporal analogue approach, this study examines 30 years of climate data (1989–2019), along with changes in operational requirements (grooming) and the behavioural responses of snowmobilers (permit sales) to record warm winter conditions across Ontario's 16 snowmobile districts. The presented research is the first empirical study to explore the spatial and temporal shifts of snowmobile activity in response to climatic conditions, providing important insights into the vulnerability and adaptive capacity of the industry in a medium (RCP4.5) and high (RCP8.5) mid-century (2046–2060) emission scenario.

## 2. Literature Review

The snowmobile industry is an important economic and cultural activity, with an estimated four million riders worldwide [22]. Given the remote location of most snowmobile trails, the recreational sport generates billions of dollars annually for rural communities, providing employment, income, and the provision of infrastructure through diverse cross-sector industries (e.g., service stations, food and beverage facilities, accommodations, retail, real estate investments, insurance agencies) [22]. Almost half of the global snowmobile market share is in North America (1.27 million riders in the US and 618,000 in Canada), with an annual economic impact of $35.3 billion, supporting 100,000 full time jobs, across an estimated 124,000 km of signed and maintained snowmobile trails [22].

The importance of climate-related conditions for snowmobiling cannot be understated, with the snowmobile experience inextricably linked to both snow quantity and quality. In an online survey of snowmobilers in Vermont (*n* = 1450), over 80% of respondents indicated that consistent snow cover along the trail adds a great deal to the snowmobiling experience, with trail connectivity (i.e., a network of snow-covered trails) considered the most important factor for snowmobiling [7]. Over 97% of respondents in an Ontario snowmobile study (*n* = 1948) indicated that they check the weather forecast when planning a snowmobile trip, with the majority stating that forecasted rain and/or freezing rain would subsequently deter them from snowmobiling (>75%), followed by warm temperatures (61%) [6].

Similar to ski tourism, the majority of studies that examine the impact of climate change on the snowmobile industry utilize models to examine operator vulnerability outcomes under different emission scenarios. The first study was published by [19], combining a snow depth model with downscaled global climate model (GCM) scenarios. The study projected that the average season length for snowmobiling in Ontario and Quebec would decrease 11% to 44% by the 2020s under a low- (B2) emission scenario, 39% to 68% under the high-(A1) emission scenario, with a reliable snowmobile season eliminated by the 2050s under the high-emission scenario. In a follow up study, Scott et al., 2008 [23] explored the US Northeast, projecting that under a high-(A1) emissions scenario, some regions would lose more than half of the current snowmobile season, and that a reliable snowmobile season (>50 days with ≥15 cm of snow) would be eliminated by the 2080s. Using temperature-indexed snow melt and accumulation equations with downscaled temperature and precipitation data, ref. [24] modeled the expected number of days with

adequate snow coverage (>10 cm) in Yellowstone National Park for over-snow vehicles (i.e., snowmobiles and snowcoaches). Under a medium-(RCP4.5) and high-(RCP8.5) emissions scenario, season length is projected to decrease 13% and 16% by 2050s, and 16% and 27% by 2080s, with the authors noting significant differences across the park (e.g., a decrease of 70% at the West Entrance Road by the 2080s). Using a variable infiltration capacity (VIC) model, ref. [25] explored changes in snow conditions for eight US states in the Great Lakes and the Midwest region. Under an RCP4.5 and RCP8.5 scenario, respectively, snowmobiling days are projected to decrease by 21 and 23 in the 2020s, 21 and 47 in the 2050s, and 38 to 62 days in the 2080s, with zero snowmobiling days projected by the 2080s in the southern-most regions under both emissions scenarios. Simulating natural snow accumulation at 247 winter recreation areas across the US, ref. [26] projects changes in the average season length for snowmobiling to range from small increases to significant declines (>80%) under the RCP4.5 scenarios in 2050, with inter-annual variability remaining high in the 2050s and effectively collapsing by late century under an RCP8.5 scenario (i.e., larger fraction of snowmobile locations to have season length reductions of >80%). The authors note that the spatial distribution of impacts differs considerably across the US, with higher elevations (Rocky Mountains and Sierras) less vulnerable compared to lower elevation locations (upper Midwest and New England). When the modeling results were monetized using current prices for US snowmobile permits (accounting for population change), the study estimates that the shortened snowmobile season under RCP8.5 could amount to an economic loss of $1.44 million (or an 11% decrease in snowmobiler visits in the 2050s) to $5.47 million (a 43% decrease in the 2090s).

Although growing, research on the behavioural response of tourists to climatic change continues to lag behind modeling-based approaches [21], with a near absence of research on how snowmobilers will (or have been) respond(ing) to the seasonal variability projected by the aforementioned model-based studies. In Vermont, of the 45% of surveyed snowmobilers who have noticed a shortened winter season and the 38% who have noticed a decline in snow depth, the majority (74% and 62%, respectively) have subsequently decreased the amount they snowmobile [7]. Based on the survey responses, the authors estimate that a season with ≤79 days to snowmobile would result in a statistically significant decrease in the sport, particularly once climate-related changes become more pronounced (i.e., noticeable) to snowmobilers in the Vermont winter landscape. Using the same survey, ref. [8] examined the climate sensitivity of local snowmobilers compared to non-locals, with the former more likely to reduce the amount they snowmobile when trail network conditions change, such as when trail connectivity decreases, or if encounters with other snowmobilers increase (e.g., as users concentrate on a more limited network). With only one known survey, the presented research adds much needed insight into the behavioural response of snowmobilers to projected climate change, offering empirical insight into the spatial and temporal shifts in snowmobiler activity in response to warming temperatures.

## 3. Method

Canada's climate is warming at a rate about twice that of the global average, with mean annual temperature and precipitation increases in Ontario of 1.3 °C and 9.7% between 1948 and 2016, respectively [27]. Consistent with an increase in mean temperature, extreme cold temperatures have become less cold, duration of snow and ice cover is decreasing, including significant reductions in seasonal snow accumulation projected to mid-century [27]. Due to Ontario's vast size and diverse topography, the impact of climate change will vary. The province has three main climatic regions, listed here from the mildest to the coldest winter season: (1) a hot-summer humid continental climate (Koppen *Dfa*) in parts of southwestern Ontario and the Niagara region, (2) a warm-summer humid continental climate (Koppen *Dfb*) which covers central and eastern Ontario, and (3) a subarctic climate (Koppen *Dfc*) which extends across the northernmost points of Ontario (north of 50° N) [28].

A major driver of Ontario's winter economy, snowmobiling is estimated to contribute $3.3 billion ($403.9 million in direct gross domestic product [GDP]), supporting over

6400 jobs across the province [29]. With 155,000 registered snowmobiles, the province represents 25% of the Canadian market [22]. The trail network is vast, covering distances of over 34,000 km. The network is maintained under the leadership of the Ontario Federation of Snowmobile Clubs (OFSC), which is a volunteer-led not-for-profit association representing the 16 provincial snowmobile districts, which includes its 183-member snowmobile clubs [30]. All riders are required to purchase a permit prior to entering a trail for recreational riding purposes. Snowmobilers are encouraged to purchase a permit for the district they will be riding in, as sales directly support the maintenance of the trail system. OFSC notes evidence of reduced snowmobile activity during poor weather conditions, resulting in decreased trip activity and a subsequent decrease in expenditures and associated economic activity [29]. The overall goal of this study is to understand the differential climatic impacts and adaptive responses of snowmobilers across the 16 regional snowmobile districts in Ontario (Figure 1).

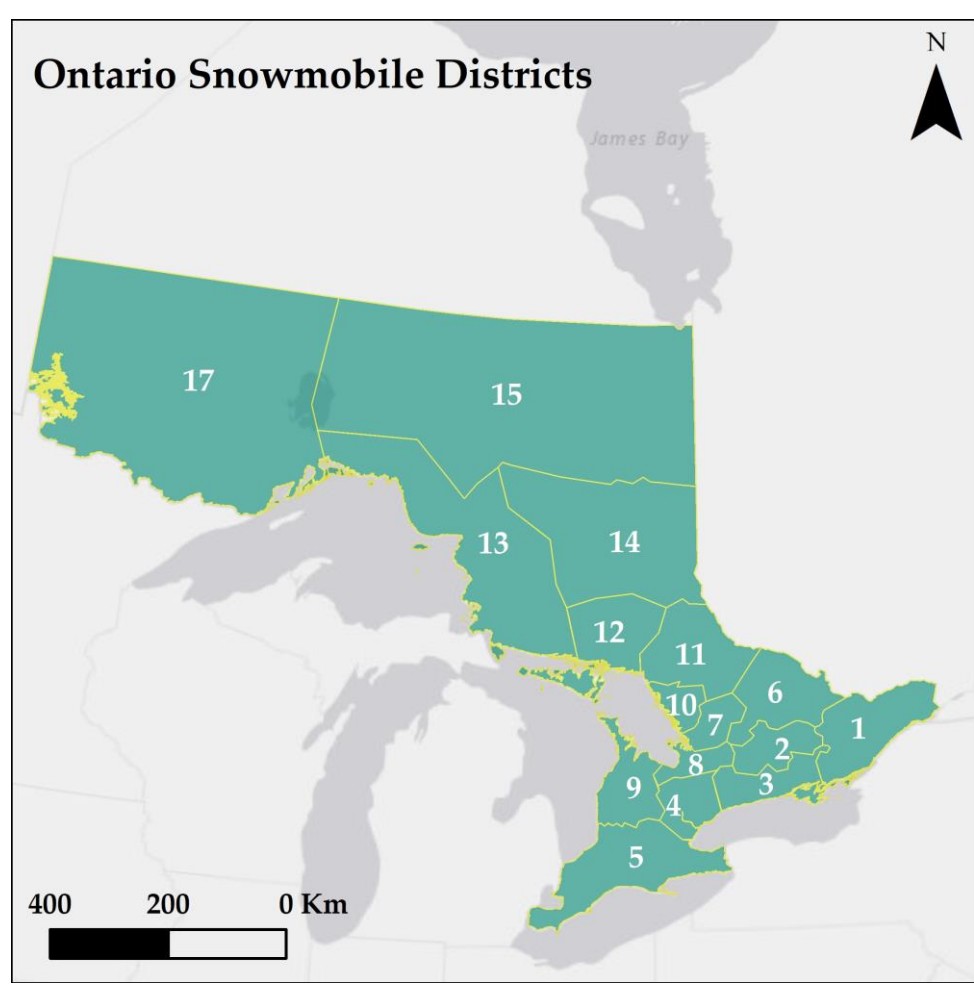

**Figure 1.** Ontario Snowmobile Districts, numbered in accordance with the OFSC trail network. (Adapted from [30]).

To identify the climate baseline (or climate normal) for each of the 16 Ontario snowmobile districts, a geospatially representative weather station in each district was selected based on both proximity to trail locations and data availability as follows: District 1—Brockville, District 2—Peterborough/Trent University, District 3—Trenton, District 4—Georgetown, District 5—Woodstock, District 6—Petawawa, District 7—Muskoka, District 8—Shanty Bay, District 9—Wiarton, District 10—Parry Sound, District 11—North Bay, District 12—Sudbury, District 13—Sault Ste Marie, District 14—Timmins, District 15—Kapuskasing, and District 17—Sioux Lookout (Figure 1). Thirty years of weather data

(1989–2019) for the winter snowmobiling season (November–April) was obtained from the Government of Canada's Historical Climate Data database (https://climate.weather.gc.ca/index_e.html, accessed on 1 August 2020) and https://climatedata.ca (accessed on 1 August 2020), which is a collaboration between the Canadian Government (Environment and Climate Change Canada), the Computer Research Institute of Montreal, Ouranos, the Pacific Climate Impacts Consortium, the Prairie Climate Centre, and HabitatSeven. Monthly data was downloaded; where monthly data was not available, daily weather was downloaded and monthly averages were calculated. The climate variables included mean temperature, mean maximum temperature, mean minimum temperature, precipitation total, total snow, and snow on ground.

Next, high-resolution climate change projection data through the Ontario Climate Data Portal (OCDP) was obtained for the mid-twenty-first century warming years (2046–2060) for an RCP4.5 scenario (i.e., representative of a medium greenhouse gas emission [GHG] future) and an RCP8.5 scenario (i.e., representative of a high-emission future). Regional climate models (RCMs) were selected based on the geographic coordinates of each of the 16 weather stations selected. Monthly and seasonal averages (i.e., November–March) were then calculated for each district and for each weather variable in the RCP4.5 and RCP8.5 projections, as well as for the baseline period (i.e., the 30-year seasonal average from 1989 to 2019). Both temperature (mean, mean maximum, mean minimum) and precipitation (total precipitation, total snow, snow on ground) variables were downloaded. However, due to significant gaps and data inconsistencies with respect to precipitation variables, particularly in northern districts, temperature variables were primarily relied on when selecting representative seasons (i.e., within 1° of the baseline, RCP4.5 and RCP8.5). From this analysis, the 2017–2018 snowmobiling season was selected as the baseline season, 2016–2017 was selected as representative of a mid-century (2040–2060s) RCP4.5 scenario, and the 2015–2016 winter season was selected as representative of the mid-century RCP8.5 scenario. It is important to note that analogue years temporally close to the baseline year were chosen to minimize the influence of other social or economic factors that could contribute to industry or snowmobile use patterns.

Operational (i.e., grooming hours) and behavioural (i.e., permit sales) data was then examined at a provincial and district level for the baseline period, and the two mid-century climate scenarios (RCP4.5, RCP8.5). Grooming hours were selected because trail grooming is typically the single greatest expense facing the operation of a snowmobile trail system in terms of capital costs (e.g., machine purchase) and ongoing operational costs (e.g., fuel, labour). Moreover, warm temperatures ($\geq -5\,^\circ$C) and the resultant warm, wet, or saturated snow reduces the efficiency of the grooming process, making it more energy intensive (i.e., it takes more hours to groom trails) [31,32]). Permit sales are a direct response of snowmobilers to the seasonal conditions, with sales utilized as a key metric in understanding consumer response to climatic conditions [33]. Permit sales data included (1) seasonal permits (2023 season: $280 CAD, for individuals with a sled model year 2000 or newer), (2) classic permit (2023 season: $190 CAD, for individuals with a sled model year 1999 and earlier), and (3) daily permit (2023 season: $45 CAD, can only be purchased for a minimum of two consecutive days, and only between December–March of each season). Due to the proprietary nature of the operational and sales data, only the percentage differences between the baseline and RCP4.5 and RCP8.5 seasons have been included in the results. While access to more detailed data would further enhance our understanding of the limits to adaptation and long-term business sustainability, as the first study to utilize industry data in response to climatic conditions, the empirical insights are nevertheless insightful for informing risk assessments in Ontario and other major snowmobile markets (e.g., Canada, Europe, USA). The findings from this research are presented at both a provincial level, as well as stratified by the 16 individual districts to assess the spatial differences in climatic risk across the trail network.

## 4. Results

Across the 16 Ontario snowmobiling districts, the average winter temperature (Tmean) from November to March for the 30-year period (1989–2019) was −5.5 °C, with the lowest seasonal average in District 15 (−11.5 °C) and the highest in District 5 (−1.7 °C) (Figure 2). While multiple temperature and precipitation variables were assessed when selecting analogue years, due to inconsistencies in data availability across districts (particularly in northern districts), mean temperature was most relied on. The 2017–2018 snowmobiling season was then selected as the baseline season, which most closely represents the average seasonal temperature of the 30-year period. Next, based on regional climate projections, the 2016–2017 was selected as representative of a mid-century (2040–2060s) RCP4.5 scenario, with a seasonal Tmean of −3.4 °C, which is +3.1 °C from the baseline. The 2015–2016 winter season was selected as representative of the mid-century RCP8.5 scenario, with provincial Tmean of −2.7 °C, an increase of +3.8 °C from the baseline period. While the average seasonal temperature increased in all districts in both the 2015–2016 and 2016–2017 season, Districts 10 and 17 experienced the largest increase in Tmean (Table 1).

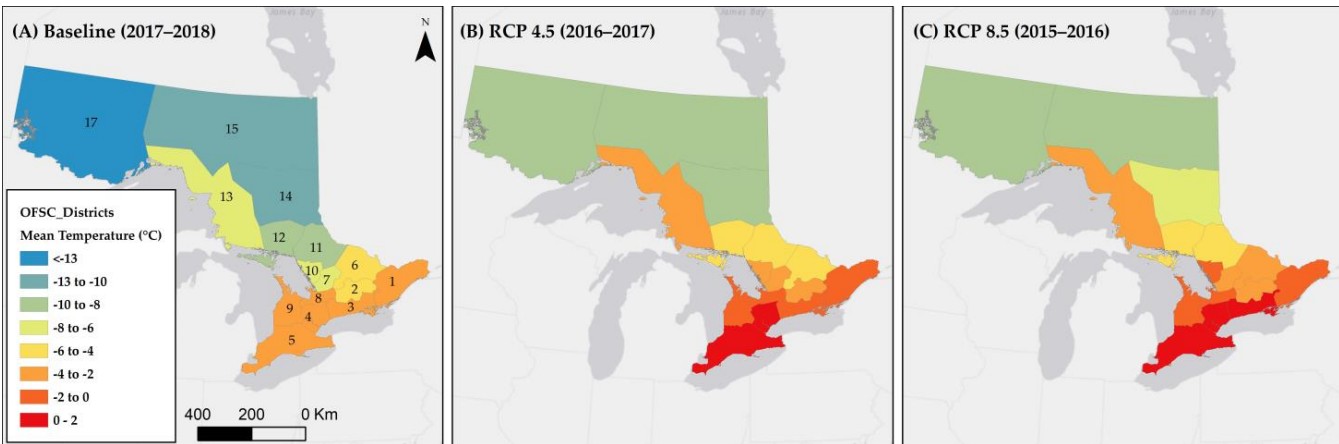

**Figure 2.** Average temperature for November to March for the following scenarios: (**A**) the climatically normal baseline for the 30–year period (1989–2019), represented by 2017–2018; (**B**) RCP4.5, represented by 2016–2017 with a seasonal mean of −3.4 °C (3.1 °C warmer than the baseline average); and (**C**) RCP8.5, represented by 2015–2016 with a seasonal mean of −2.7 °C (3.8 °C warmer than baseline). The symbology for panel A is applied to all panels.

Table 2 shows the relative change in grooming hours for the RCP4.5 and RCP8.5 analogue years with the intra-year proportion of total grooming hours for each district. Large percentile changes are less significant if the proportion of the total is small (e.g., <5%). Compared to a climatically normal winter season (i.e., 2017–2018 baseline), grooming hours in the RCP4.5 season increased by 8% across the province, with 10 of the 16 districts requiring more grooming hours during the warmer season. The greatest increase was in District 6 (+91%), which accounted for 16% of the total provincial grooming hours, followed by Districts 12 (+53%), 4 (+28%), and 5 (+23%). Conversely, over one third of the districts saw a decrease in grooming hours, including Districts 3 (−31%), 14 (−38%), 15 (−30%) and 17 (−25%), constituting 18% of the total grooming hours in the province. Relative to the baseline year, the total number of grooming hours decreased by 28% in the RCP8.5 season. The largest reductions in grooming hours were in Districts 3 (−68%), 4 (−66%), and 6 (−58%) with marginal increases in grooming hours observed in Districts 5 (+0.4%) and 11 (+5%) (Figure 3 and Table 2). In terms of the proportion of grooming hours distributed across Ontario, the grooming hours remain consistent across most districts, with the exception of marked decreases in District 6 (16% in RCP4.5 to 5% in RCP8.5), and increases in District 14, 15 and 17 (+4%, 4%, +3%, respectively). The increase in proportion noticed in the northern most districts is in spatial agreement with the distribution of the coldest

average winter temperatures in both RCP4.5 and RCP8.5 scenarios. The relationship between average grooming hours and temperature among districts is weakly negative (Spearman's rank order correlation) of $r = -0.21$ and not statistically significant ($p = 0.43$). This is somewhat expected, as the variance in size and spatial distribution of Districts causes clustering in the temperature data.

**Table 1.** Average seasonal temperature by Ontario snowmobile districts, representative of a climatically normal (baseline), medium-(RCP4.5) and high-(RCP8.5) GHG emission scenario.

| District | Tmean (°C) November–March | | |
| --- | --- | --- | --- |
| | Baseline (2017–2018) | RCP4.5 (2016–2017) | RCP8.5 (2015–2016) |
| 1 | −3.2 | −0.9 | 0.0 |
| 2 | −4.3 | −2.0 | −2.4 |
| 3 | −3.0 | −0.9 | 0.4 |
| 4 | −2.3 | 0.3 | 0.9 |
| 5 | −2.0 | 0.6 | 2.0 |
| 6 | −5.9 | −4.2 | −3.0 |
| 7 | −6.3 | −3.8 | −2.4 |
| 8 | −3.6 | −1.1 | −0.2 |
| 9 | −3.3 | −0.6 | 0.0 |
| 10 | −6.6 | −2.0 | −1.2 |
| 11 | −8.6 | −5.9 | −4.5 |
| 12 | −8.6 | −5.6 | −5.2 |
| 13 | −6.9 | −3.0 | −2.2 |
| 14 | −12.4 | −8.2 | −7.8 |
| 15 | −13.2 | −8.6 | −8.6 |
| 17 | −13.5 | −8.5 | −8.3 |
| Ontario | −6.5 | −3.4 | −2.7 |

**Table 2.** Change in mean grooming hours between a climatically normal (baseline) season and a representative medium-(RCP4.5) and high-(RCP8.5) GHG emissions season across Ontario snowmobile districts.

| District | Change in Grooming Hours (%) | | | |
| --- | --- | --- | --- | --- |
| | RCP4.5 (2016–2017) | RCP4.5 Proportion of Total | RCP8.5 (2015–2016) | RCP8.5 Proportion of Total |
| 15 | −30 | 8 | −27 | 12 |
| 14 | −38 | 5 | −25 | 9 |
| 6 | 91 | 16 | −58 | 5 |
| 1 | 13 | 9 | −24 | 9 |
| 8 | 15 | 9 | −38 | 7 |
| 9 | −5 | 7 | −32 | 7 |
| 11 | 26 | 9 | 5 | 11 |
| 13 | −5 | 6 | −35 | 6 |
| 12 | 53 | 8 | −27 | 6 |
| 7 | 10 | 5 | −16 | 6 |
| 17 | −25 | 3 | −15 | 6 |
| 2 | 28 | 5 | −8 | 5 |
| 10 | 8 | 4 | −9 | 5 |
| 3 | −31 | 2 | −68 | 2 |
| 5 | 23 | 2 | 0 | 3 |
| 4 | 28 | 2 | −66 | 1 |
| Ontario | 8 | 100 | −28 | 100 |

Note: Districts are listed in order of total grooming hours in the baseline year.

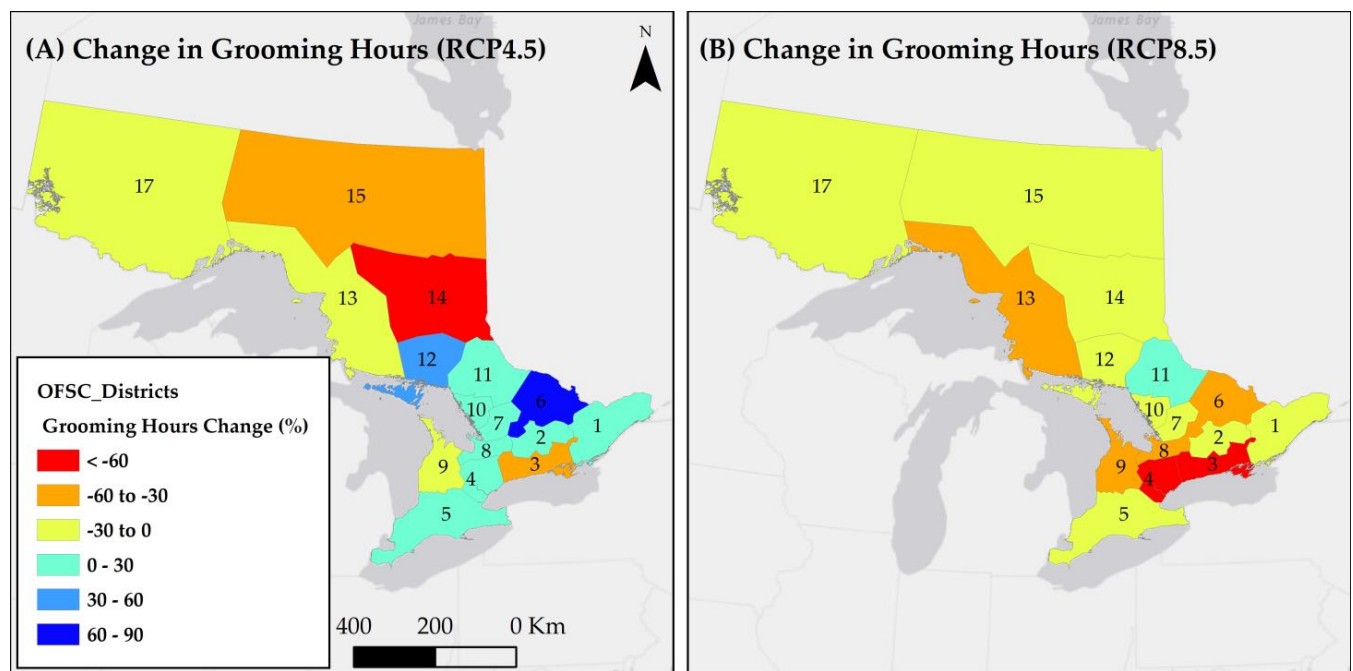

**Figure 3.** Distribution of the percent change in grooming hours per district between a climatically normal (2017–2018) winter season and an RCP 4.5 (**A**), (**left**) and RCP 8.5 (**B**), (**right**) scenario.

The total number of permit sales in Ontario decreased in the RCP4.5 representative season (−2%) compared with the baseline season, with noticeable variability based on permit type and district location (Figure 4). Seasonal permit sales in Ontario decreased by 0.6% overall, with the greatest decreases in seasonal permits in Districts 6 (−19%), 13 (−7%) and 15 (−8%), respectively. Importantly, the majority of districts experienced an increase in seasonal permit sales, including a 9% increase in District 1, +5% in District 5, +4% in Districts 4 and 17, and +1–2% in Districts 3, 7, 9, 10 and 12. For classic permit sales, there was a substantial increase across Ontario (+19%) compared to the baseline season, with all districts experiencing an increase ranging from +2% in District 2 to +36% in District 4. Total daily permit sales decreased in the RCP4.5 season compared to the baseline year (−30%), with only District 1 recording an increase in daily permit sales (+11%). The most adversely impacted Districts in terms of daily permit sales were Districts 5 (−59%), 9 (−49%), 12 (−42%), 14 (−48%) and 17 (−66%).

Compared to a climatically normal season, total permit sales decreased −5% during the RCP 8.5 winter season. Similar to the RCP4.5 season, changes in permit sales differed based on permit type and district location (Figure 5). The decrease in seasonal permit sales was much more pronounced in the RCP8.5 season, decreasing by 8% relative to the baseline. The greatest drop in sales occurred in Districts 6 (−32%), 13 (−20%) and 15 (−21%), respectively. While ten districts experienced an increase in seasonal permit sales during the RCP 4.5 season, only two districts had an increase in seasonal permit sales in the RCP8.5 season, including District 4 (+6%) and District 5 (+4%). Similar to the RCP4.5 season, classic permit sales had an overall increase of +19% across the province in the RCP8.5 season, with all districts increasing permit sales compared to the baseline year, the largest of which include Districts 1 (+27%), 4 (+45%), and 12 (+36%). Conversely, there was a decrease in total daily permit sales (−9%), with the greatest losses recorded in Districts 5 (−44%), 13 (−42%) and 17 (−42%). However, not all districts experienced a loss of daily permit sales in the RCP8.5 season, with significant increases in Districts 1 (+76%), 4 (+22%), 6 (+53%), 7 (+15%), 11 (+45%) and 12 (+29%).

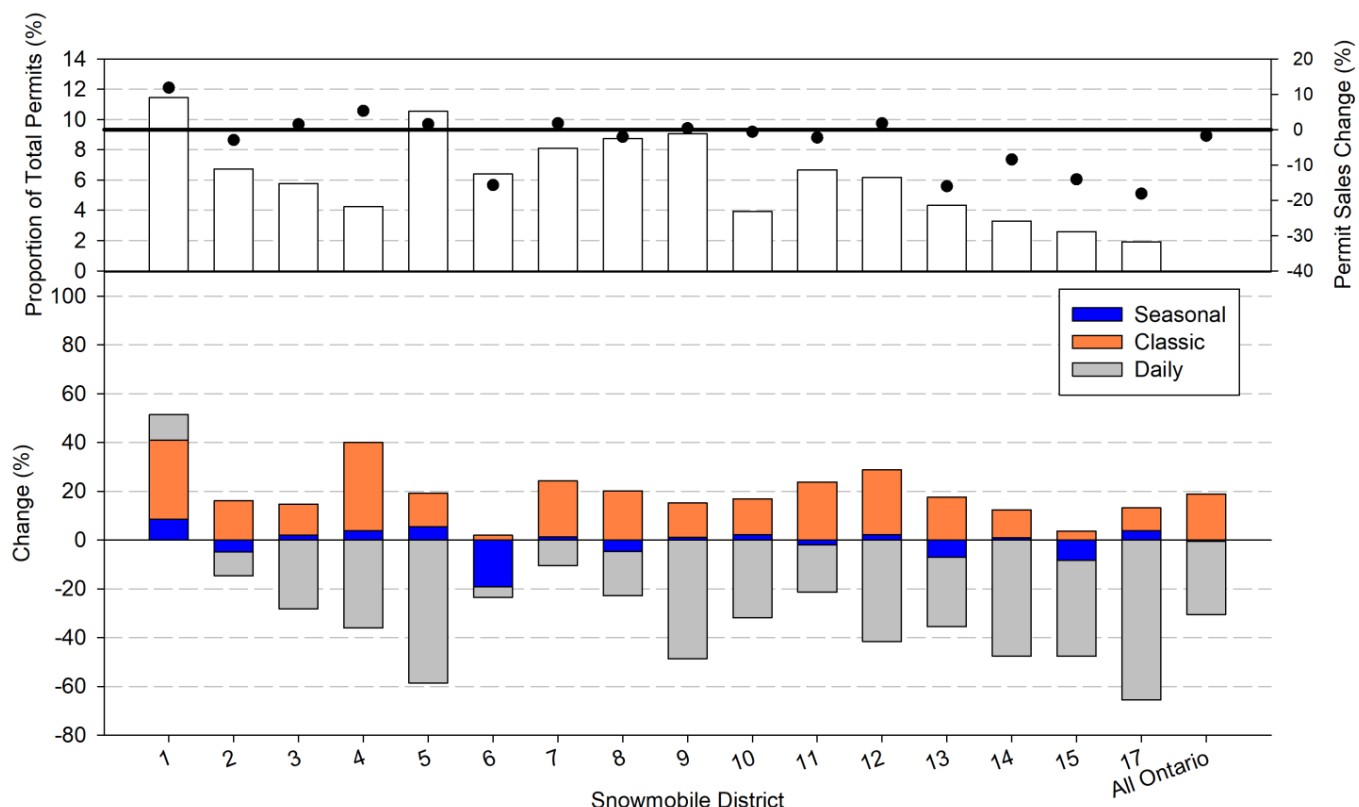

**Figure 4. Top panel**: Proportion of total permits sold across Ontario snowmobile districts for the RCP4.5 (2016–2017) winter season represented as a bar graph. Dots on the graph (right axis) indicate the percent change in total permit sales between the baseline and the RCP4.5 analogue with the bold line representing no change in permit sales. **Bottom panel**: percentage change in sales by permit type between a climatically normal baseline (2017–2018) and the RCP4.5 (2016–2017) winter season across Ontario snowmobile districts.

　　In terms of the proportion of total permit sales, between the RCP4.5 (Figure 4) and RCP8.5 (Figure 5) scenario, the change in the proportion of total permit sales for each district was less than 1% across all districts ($R^2$ = 0.994, *p* = <0.001) (i.e., the additional warming present during the 2015–2016 compared to 2016–2017 had little to no additional impact on the relative distribution of permit sales across the province). With a Komolgorov–Smirnov statistic of 0.125 and a *p*-value of 0.999, the findings indicate that the district permit sales follow the same distribution across the baseline, RCP4.5 and RCP 8.5 analogues (i.e., Districts 1 and 5 constitute approximately 10% of overall permit sales for each respective season). Importantly, the distribution of sales is not equal across each type of permit, with seasonal permit sales representing 78% of overall permits, followed by 11% for both classic and daily permits in the baseline year. The relative magnitude of permit types is important to consider when observing the relative increase/decrease in permit sales type, because large increases in classic and daily permit sales can still result in an overall decrease if seasonal permits purchases are negative (e.g., District 11 in Figures 4 and 5).

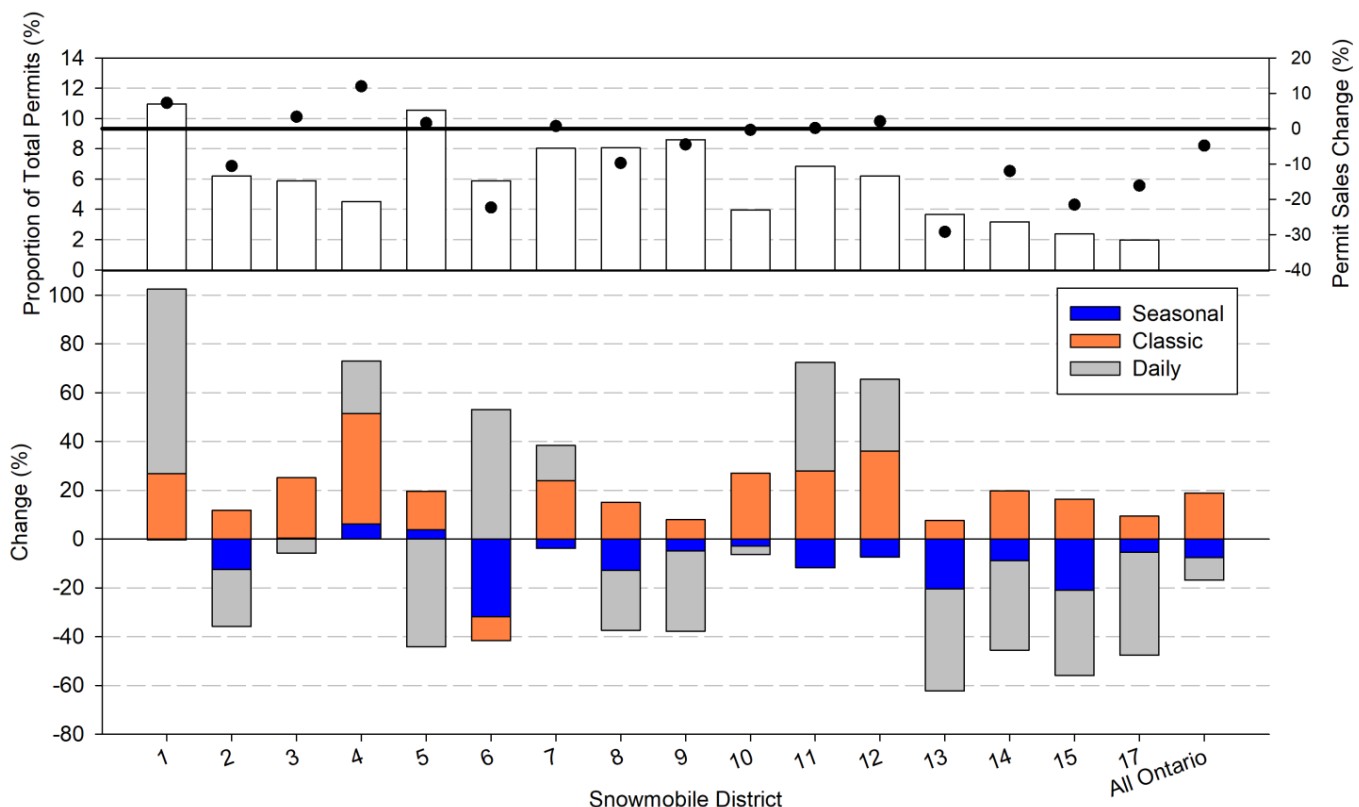

**Figure 5. Top panel**: Proportion of total permits sold across Ontario snowmobile districts for the RCP8.5 (2015–2016) winter season represented as a bar graph. Dots on the graph (right axis) indicate the percent change in total permit sales between the baseline and the RCP8.5 analogue with the bold line representing no change in permit sales. **Bottom panel**: Percentage change in permit sales between a climatically normal (2017–2018) and RCP8.5 (2015–2016) winter season across Ontario snowmobile districts.

## 5. Discussion

The spatial patterns of snowmobile trail use and maintenance are largely unknown, with a virtual absence in the academic literature with respect to how the industry responds to marginal conditions. Through an examination of grooming data, it is evident that trail maintenance increases substantially by mid-century in the RCP4.5 scenario, as mean seasonal temperatures rise +3 °C from the baseline (−7 °C to −4 °C, respectively), requiring more grooming hours (+8% overall) to attend to increased climatic variability, including more incidences of snowmelt and rain. A decrease in permit sales would be particularly costly in those districts with large increases in grooming hours (e.g., by 91% in District 6 and 53% in District 12). The few exceptions are in districts located in the north of the province (14, 15, 17) that experienced a decrease in grooming hours (≤25%), which is understandable given that mean temperatures will continue to be climatically favourable for snowmobiling in those regions (≤−8 °C). As temperatures continue to rise in the RCP8.5 scenario, grooming hours significantly decrease in response, particularly in the southern districts, with fewer trails requiring maintenance due to higher temperatures and lower natural snowfall (i.e., a shorter winter season). A decrease in grooming would yield some operational savings by reducing hourly labour and fuel costs, but consolidating groomers (e.g., multiple districts sharing equipment) is unlikely to be an option in Ontario given the vast trail distances across each individual district.

In terms of snowmobiler response to warming conditions, changes in permit sales underscore important temporal and spatial variability. Not only are there key differences in terms of how snowmobilers responded across each district (e.g., some districts experienced an increase in sales while others faced decreased sales), but important differences were also

recorded based on permit type. In both analogue seasons, classic permit sales increased 19% across Ontario, with 12 of the 13 districts recording an increase in sales, indicating that riders with classic sleds (i.e., manufactured in or before 1999) are highly resilient to climatic variability. In terms of seasonal permits, sales remained somewhat stable in the RCP4.5 season (−1% overall), whereas daily permit sales decreased across all Districts (by 30% overall) except District 1 (+11%). It is particularly interesting is that in the RCP8.5 season, while seasonal permits continued to decline (by 8% overall), daily permit sales were highly sensitive, with either a notable increase or decrease in sales (e.g., 13 of the 16 districts experienced a ≥22% change in daily permit sales compared to the baseline season). These findings suggest that there could be an increase in market share for some districts (e.g., Districts 1, 6, 7, 11, 12), with snowmobilers adapting by purchasing daily permits in lieu of seasonal permits to take advantage of climatic conditions when and where they are available. In other words, despite projected decreases in season length [19,25,26], snowmobilers are willing to adapt and unwilling to give up the sport. Modeling-based studies that only consider season length may therefore be overestimating the impact of climate change on the snowmobile industry. While a shortened season certainly has implications for permit sales, it does not preclude snowmobiling altogether, with similar substitution behaviours recorded in the ski literature (i.e., low rates of activity substitution in seasons with poor snow conditions) [34–37]. The findings from this study also support [8], whereby snowmobilers in Vermont sought out trails in marginal seasons. It was noted that Vermont snowmobilers adapted differently, such that local snowmobilers were more resilient versus non-local snowmobilers. Further research into behavioural adaptation based on segments would be valuable, such as the degree to which substitution behaviours vary based on place of residence (primary and secondary), age, experience, and district (or club) loyalty. Such research would also yield important insight into the spatial and temporal shift from seasonal permits to daily and classic permits, including why snowmobilers with classic sleds are more resilient to marginal conditions.

As snowmobilers adapt spatially (e.g., travel to climatically viable districts) and temporally (e.g., purchase daily permits versus seasonal permits), riders may begin to concentrate along a smaller network of trails. Much like climate-related conditions are important when deciding whether to go snowmobiling [6], encounters with other snowmobilers directly influence the quality of the recreational experience [7], with the ability to travel great distances, including the interconnectivity of trails, deemed vital to the snowmobiling experience [8]. It remains unclear whether and to what degree crowding and trail fragmentation will remain (un)acceptable as trail networks truncate over time, with investments in additional trail networks and associated infrastructure (e.g., groomers, service stations) in climatically viable districts an important consideration moving forward. Based on geographic size, temperature projections, and total number of permits sold, Districts 1 and 5 may be at greatest risk of crowding. However, it is important to underscore that future research is needed to model snow and ice presence and thickness to better assess trail viability, recognizing that some trails within a district are more or less climatically vulnerable than others (e.g., trails that traverse through a forest will maintain a longer snow season versus trails that transect rivers and lakes).

Another important consideration is the economic implications of snowmobilers shifting from seasonal to daily permits. Seasonal permits not only generate higher revenues, but they are important for supporting the upfront costs faced by the snowmobile clubs as they prepare the trails for the season. As climatic variability increases in the future, more snowmobilers may continue to shift away from seasonal permits to daily permits. Following the COVID-19 pandemic when provincial trail operations were ceased, a "rider advantage confidence benefit" was promoted to encourage Ontario snowmobilers to purchase a permit well in advance of the winter season (e.g., $50 bonus applied towards the permit if purchased before 30 September). Similar discounts or other financial incentives may be necessary to support seasonal permit sales in the future or conversely, a fee increase for daily permits to offset reduced revenues. An exploration into the implications

of various fee structures on behavioural adaptation would be an interesting avenue for future research.

## 6. Conclusions

Global climate mitigation policy will determine our emissions pathway and play a decisive role in the future viability of winter tourism, including snowmobiling. Regrettably, climate change has yet to become a priority for tourism policy makers [38], with no known national nor regional climate policy that includes content on tourism in North America [2]. There is an opportunity for the snowmobile industry to engage in climate action policies not only to limit adverse impacts of warming, but to advance the required transition to a low carbon society. The future of snowmobiles is expected to be shaped by rising consumer concerns about vehicular emissions, with electric snowmobiles anticipated to gain rapid popularity (e.g., projected valuation of USD 5 billion by 2032) [39]. Electric sleds may also encourage a new demographic of recreational users, as they provide a quieter and more sustainable alternative to conventional gasoline-powered snowmobiles.

Irrespective of mitigation efforts, the IPCC is explicit that climate impacts in the coming years and decades are unavoidable and worsening, which will indisputably alter the competitiveness of destinations and reshape tourism demand [1,15]. Tourists' responses to climate continue to be a critical yet understudied component of the tourism system, with an often-deterministic assumption that a change in a climatic variable (e.g., increase in temperature), will automatically elicit a corresponding shift in demand (e.g., decrease in demand) [2]. The findings from this research underscore the need to move beyond season length as the primary indicator of industry vulnerability, with evidence that snowmobilers are willing to adapt spatially and temporally to pursue the sport. Snowmobile networks are also more nuanced than what is projected by model-based studies, which fail to capture the current operating reality of partial network capacity (i.e., some trails are open longer within a network or district).

Given that neither snowmobilers nor the industry responds to climatic changes in isolation, a systems-based approach is an important next step to assess the synchronous evolution of demand and supply changes in a warming world. Integrated assessments that couple changes in snowmobile seasons, including varying impacts across trail networks, with associated shifts in demand (substitution) are needed to explore regional markets (e.g., Ontario) and individual destinations (e.g., districts). Much like other winter-sports tourism, climatic vulnerability will not be uniform across the snowmobile industry. The continued ways in which snowmobilers will adapt their recreation behaviour in response to changing trail networks, including the growth of electric sleds and associated network of charging infrastructure, warrants further attention.

**Author Contributions:** Conceptualization, M.R.; methodology, M.R., F.C. and G.G.; validation, M.R., F.C. and G.G.; formal analysis, M.R. and F.C.; data curation, F.C.; writing—original draft preparation, M.R. and F.C.; writing—review and editing, M.R., F.C. and G.G.; visualization, G.G.; supervision, M.R.; project administration, M.R.; funding acquisition, M.R.; All authors have read and agreed to the published version of the manuscript.

**Funding:** This research was funded by MITACS through the MITACS Accelerate Fellowship #T17682 in conjunction with the Ontario Federation of Snowmobile Clubs to Michelle Rutty and Francesca Cardwell.

**Institutional Review Board Statement:** Not applicable.

**Informed Consent Statement:** Not applicable.

**Data Availability Statement:** Value added data is available on request to the authors due to privacy concerns. Raw data is the property of the Ontario Federation of Snowmobile Clubs and privately held.

**Conflicts of Interest:** The authors declare no conflict of interest.

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
