# Peer review of "Snowmobiling and Climate Change: Exploring Shifts in Snowmobile Activity Using a Temporal Analogue Approach in Ontario (Canada)"

_tourismhosp, doi:10.3390/tourhosp4040037_

Round 1
Reviewer 1 Report
Comments and Suggestions for Authors
Taking the Ontario (Canada) for example, this paper explored the vulnerability and adaptive capacity of snowmobile industry in a medium (RCP4.5) and high (RCP8.5) mid-century (2046-2060) emission scenario. Overall, although authors have done some work, there exist some flaws that need to be revised. My major concerns are as follows.
1) The research topic of this paper and its corresponding scientific questions are not clear enough. Specifically, the title of this paper is Snowmobiling and Climate Change. However, this paper mainly focused on this paper explored the vulnerability and adaptive capacity of snowmobile industry in different emission scenario. Moreover, based on the scientific problems of this paper, the author needs to clearly state in the first two paragraphs of the introduction what is the concern of this paper? However, the author does not provide that.
2)Regarding Literature Review, the author mainly presents the existing research progress in a list way, but lacks a summary of the existing literature. At the same time, in view of these shortcomings existing in the existing research, what improvements or supplements have been made in this paper? What is the marginal contribution of this paper? None of this is clear.
3)Regarding the policy implications of this paper, This paper does not pay due attention to it. It is suggested that the authors supplement the policy implications of this study.
Comments on the Quality of English Language
Good!
Reviewer 2 Report
Comments and Suggestions for Authors
Overall Comment: Well written interesting manuscript that uses analog years to infer the impact of climate change on snowmobiling.
Main Comments
More information is needed as to why 2017-2018, 2016-2017 and 2015-2016 were chosen as the base-, RCP4.5, and RCP8.5 years. See Table 1 comments below.
Colorful plots attract much more attention than tables. I suggest converting several of your tables to color plots overlaid on a map of Ontario. Plots would also make it easier to infer how changes in baseline temperature in a district affect the response of grooming to increases in temperature.
General Comments
L172-176: What is the temporal resolution of these data sets? Monthly? Seasonal? Daily?
L193: as temporally close ïƒ temporally close
L206: Is it possible to give the total number of seasonal permits, classic permits, and daily permits for Ontario as a whole?
L234-235 Table 1: For each district, in addition to mean temperature it would be useful to include the total snowfall and the number of days with at least 10? cm of snow on the ground. This values would be useful for determining the representativeness of the baseline, RCP4.5 and RCP8.5 years and also the large decreases in grooming hours between RCP4.5 and RCP8.5. A one degree change in mean temperature doesn’t seem large enough to cause the large change.
L247-249: Table 2: Is the number of grooming hours in each District proprietary information? If no, that would be an interesting add to the Table. If yes, would it be possible to order the Districts by the total number of grooming hours.
L247-249: Table 2: Are the percent change values for all of Ontario weighted by the grooming hours of each District?
L247-249: Table 2: More discussion of the non-linear response of grooming hours to warming would be useful as 1 additional degree of warming had a huge impact on grooming hours.
L250-264. This section is tedious to read without referring to a map showing the location of each district and knowing the contribution of each permit type to total sales. Please re-write to emphasize how different the changes are between daily permits (consistently down), classic permits (consistently up), and season permits (mixed results) without getting bogged down in giving values for different districts.
L266-267: Figure 2: The relative contribution of each permit type to the overall change is impossible to determine without knowing the number of each permit sold. Is it possible to give the total number of seasonal- classic-, and daily-permits sold after aggregating to include all of Ontario?
L300: Why would people with classic sleds be more resilient to climate change? Explore this in more detail: Do people have multiple snowmobiles and bring out the older ones when conditions are marginal? Are newer sleds not rented when conditions are poor? Are the people with classic sleds older? Do they live in more rural regions of the District where conditions are better? Etc.
L318: Are permits only good for one District or can they be used cross-district?
L335: Which districts are most likely to be affected by crowding?
Minor and Grammatical Comments
Be consistent with the use of negative numbers. For example, in L94 minuses are not appropriate in front of 39% and 68% as your refer to decreases earlier in the sentence. See also L104, L105, L108, L121, L122 and perhaps others.
To improve readability, spaces before km would be useful in L54, L77 and perhaps other lines.
L112: projects the average season length ïƒ projects changes in the average season length
L144: projected through to ïƒ projected through
L154: The trail network across is vast, covering distances of overïƒ The trail network is vast covering over
L155: , which requires all riders to purchase ïƒ . All riders are required to purchase
L158: for the ïƒ of the
L234-235 Table 1: I’d suggest using two as opposed to one significant digit throughout the table.
Round 2
Reviewer 1 Report
Comments and Suggestions for Authors
The author has made some modifications, and the quality of this paper has been improved accordingly. Therefore, it can be published.
Reviewer 2 Report
Comments and Suggestions for Authors
Well written. Thoughtful responses to comments. No changes suggested.